

# Future extreme precipitation intensities based on historic events

Iris Manola[1], Bart van den Hurk[2,3], Hans De Moel[2], Jeroen Aerts[2]

[1]Meteorology and Air Quality, Department of Environmental Sciences, Wageningen University, the Netherlands
[2]InstituteforEnvironmental Studies, Vrije Universiteit (VU), Amsterdam, the Netherlands.
[3]The Royal Netherlands Meteorological Institute (KNMI), De Bilt, the Netherlands

*Correspondence to*: Iris Manola (iris.manola@wur.nl)

**Abstract.** In a warmer climate, it is expected that precipitation intensities will increase, and form a considerable risk of high impact of precipitation extremes. This study applies three methods to transform a historic extreme precipitation event in the

Netherlands to a similar event in a future warmer climate, thus compiling a 'future weather' scenario. The first method uses an observation-based non-linear relation between the hourly observed summer precipitation and the antecedent dew-point temperature (the Pi-Td relation). The second method simulates the same event by using the convective-permitting NWP model Harmonie, for both present day and future warm conditions. The third method is similar to the first method, but applies a simple linear delta transformation to the historic data by using indicators from The Royal Netherlands

Meteorological Institute (KNMI) '14 climate scenarios. A comparison of the three methods shows comparable intensity changes, ranging from below the Clausius-Clapeyron (CC) scaling to a 3 times CC increase per degree of warming. In the NWP model, the position of the events is somewhat different, due to small wind and convection changes, the intensity changes somewhat differ with time, but the total spatial area covered by heavy precipitation does not change with the temperature increase.

The Pi-Td method is simple and time-efficient, compared to numerical models. The outcome can be used directly for hydrological and climatological studies, and for impact analysis, such as flood-risk assessments.

## 1 Introduction

It is expected that climate change will increase the frequency and intensity of extreme precipitation events (e.g. Stocker et al., 2014; Pachauri et al., 2014). Different types of flooding may result from extreme precipitation. In urban environments,

extreme precipitation may lead to local-scale inundations, causing damage to houses and infrastructure within a time frame of several hours. On a larger river-basin scale, extreme rainfall over a period of days to several weeks may lead to river or flash floods, which may cause fatalities and can be catastrophic for the economy (e.g. Koks et al., 2014) and ecosystems (e.g. Knapp et al., 2008).

For the management of these risks, it is important to understand how the risk of extreme precipitation will change under future-weather conditions. Current knowledge on climate change and possible future climate scenarios are developed within



the IPCC (Pachauri et al., 2014). For regional and national applications, tailored climate-change scenarios have been developed, such as those for the Netherlands (Van den Hurk et al., 2014, henceforth 'KNMI'14'). An important element for the successful application of climate change scenarios within a local to regional context is that they are tailored towards the needs of policy makers who use them in order to assess adaptation strategies' effectiveness in reducing risk of adverse

effects, such as from flooding. Therefore, users of regional climate scenarios are increasingly involved in tailoring climate-change information, in order to ensure that climate-scenario information is comprehensible and applicable to policy making (KNMI'14).

In flood-risk management, there is a need for climate scenarios that provide information on how extreme weather events may

look like in the future (Aerts et al., 2014; Ward et al., 2014). The preferable way to obtain such information is to perform numerical (climate) model simulations that are sufficiently long to resolve the climate change trend (e.g., >30 years) and which have a sufficiently high resolution to adequately resolve important dynamical and thermodynamic interactions, such as convective processes. Currently, such long and precise model simulations are lacking (Bürger et al., 2014) due to computational and data-storage constraints. Therefore, a (combination of) climate modelling and statistical corrections are

usually employed, using shorter time series and providing projections on future climate (e.g., KNMI'14). For example, a common approach to dealing with climate change in flood-risk studies can be described as a 'delta-change' technique. In such a statistical approach, results from regional and global climate models are used to derive the (seasonal) change in precipitation characteristics, such as the wet-day frequency and the median or extreme precipitation. This change factor is subsequently applied to an observed time series or individual event in order to generate (extreme) rainfall under a changed

climate (Lenderink et al., 2007; Van Pelt et al., 2012; Räty et al., 2014). Another approach that is used to study precipitation extremes is to improve the low spatial and temporal resolution of long model simulations by means of statistical and dynamical downscaling techniques (Maraun et al., 2010). Such simulated time series can also be improved by using bias-correction methods that are derived from present-day simulations (Teutschbein et al., 2012; Bakker et al., 2014). Nevertheless, bias continues to exist and the uncertainties remain quite high.


Recently, a novel 'future weather' concept has been proposed in order to provide high-resolution information on relevant characteristics of specific future extreme events, such as the duration and intensity of heavy rainfall (Hazeleger et al., 2015). According to this concept, historically observed events are used as a reference and modified with the use of numerical weather-prediction models, so that the outcome shows how the same event would occur in a future, warmer climate.

Selection of events that have triggered concerns by for instance flood-risk managers using leads to a straightforward interpretation of the impacts of such an event under hypothetical future conditions. Hazeleger et al., (2015) used a high-resolution global-atmospheric model to simulate a future extreme weather event, by imposing future boundary conditions on historic NWP simulations. Lenderink and Attema (2015) developed future scenarios of local precipitation events by



perturbing the temperature and humidity boundary conditions of simulations in the regional models RACMO2and Harmonie, in order to mimic a 2°C warmer world.

The main aim of this paper is to compare a future weather simulation with two alternative scaling methods, of which one is developed in this study. All three methods were applied to the same case study of extreme precipitation that took place in the Netherlands on July 28, 2014. The future-weather method uses the outcome of the high-resolution numerical weather prediction model Harmonie (Seity et al., 2011). This model was forced with boundary conditions representing both the historic event and future conditions, in order to obtain information on how the event would behave in the future. The first scaling method follows a non-linear delta transformation (Lenderink & Van Meijgaard, 2008), based on the observational behavior of precipitation intensity (Pi) as a function of the dew-point temperature (Td) (henceforth 'Pi-Td relation'). The transformation was superimposed directly on the historical data assuming a future, warmer world. The second scaling-method is a simplistic linear delta-change technique, which takes results from the KNMI'14 scenarios in order to develop the future event.

This paper is organized as follows: Chapter 2 begins with a flowchart summary of the steps that were followed, and subsequently illustrates and discusses the three methods for projecting the future event. In Chapter 3, the Dutch case-study is presented from observations and simulated in the Harmonie model. Chapter 4 presents the future event for each method, firstly individually, and secondly by a quantitative and qualitative comparison. The final chapter summarizes the research and concludes this paper.

## 2. Methods

The steps that were followed in this paper are summarized in Fig. 1. Overall, three methods were used to transform an observed event (July 28, 2014, in the Netherlands) into a future event, assuming a warmer climate. The Pi-Td scaling method is based on summertime hourly radar precipitation data and hourly-observed dew-point temperature for the years 2008-2015. In order to create the future precipitation event, the Pi-Td transformation was applied on the precipitation data assuming a 2°C warmer Td. In the future weather method, the historic event was simulated using an ensemble of seven runs from the weather model Harmonie, using both the historic boundary and the initial conditions from the Ensemble Prediction System (ENS) of ECMWF (ENS, Molteni et al., 1996; Leutbecher et al., 2008) for that particular day. Subsequently, the relevant future ensemble was simulated by perturbing the initial and boundary conditions to represent a unified increase of 2°C, while maintaining the relative humidity constant. In the linear delta-change method a factor was used to perturb the event, again assuming a warming of 2°C.



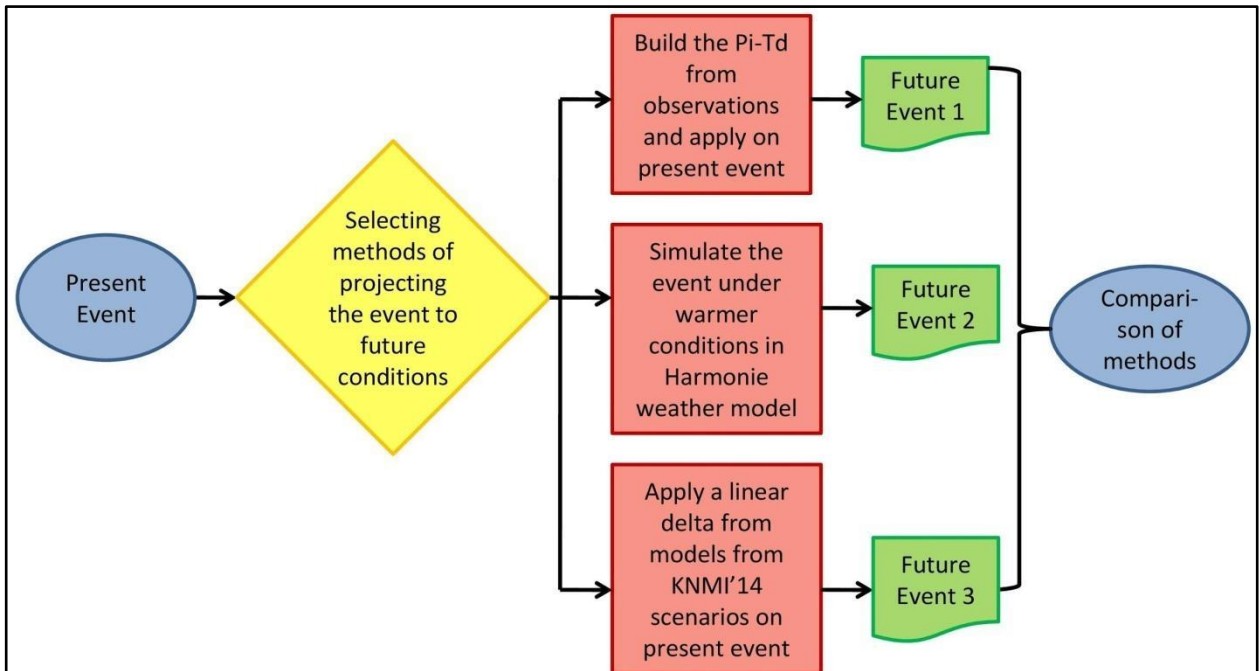

*Figure 1: In order to produce a future precipitation event, three methodologies were followed, each departing from the same historic event. A) The upper path: using observed precipitation intensity – dewpoint temperature relations in order to create change factors. B) The middle path: using the weather model Harmonie with perturbed initial conditions from ECMWF's ENS. C) The lower path: using delta-change factors retrieved from climate model simulations.*

The three methodologies were statistically compared and evaluated. As the Harmonie model has shown to be able to sufficiently simulate observed events (Attema et al., 2014; Koutroulis et al., 2015) and as it is the method that involves the highest level of physical sophistication, its outcome is used here as a benchmark for the evaluation of the Pi-Td method and the linear-delta method. The comparison between the Pi-Td method and the future simulation provides information on how the explicitly modelled interactions affect the results, compared to the statistical methods. Comparing the Pi-Td method and the linear-transformation points evaluates the added value of enhancing the sophistication of the statistical scaling approach.

## 2.1 The Dutch case study

The Netherlands is a low-lying country that is shaped by the river deltas of the Meuse and Rhine rivers. It is vulnerable to flooding from storms surging from the North Sea, as well as river flooding. In addition, extreme precipitation events inundate urban areas and agricultural fields, frequently leading to considerable damage. Observations show that the temperature in the Netherlands rose by 1.8°C since the beginning of the 20th century, clearly exceeding the global average (KNMI'14). This has led to an increase in atmospheric moisture, a 25% increase in the annual mean precipitation





and an increase of 12% per degree in the hourly intensity of the most extreme showers of the 20th century. In the KNMI'14 scenarios, the temperature is projected to rise another 1-2.3 degrees until 2050, leading to more frequent and intense extreme precipitation events.

5   The extreme precipitation event that is analyzed here took place on the July 28, 2014 and resulted in blocked highways, the disruption of air transportation, and flooded buildings and public facilities. An analysis of the 325 Dutch rain-monitoring stations shows that an event of such intensity has a 5-15 years return period (Van Oldenborgh & Lenderink, 2014). It consisted of scattered, strong convective cells that started in the early morning in the West and Southwest of the country, and reached the Central-Eastern region in the afternoon. The daily accumulated-precipitation intensity reached 140mm

10  locally (Fig. 2a). The small scale of the convective events underlines the need for high-resolution convection-permitting modelling. As the most severe damages are usually reported over urban areas (Ward et al., 2013), this analysis mainly focusses on the period between 8am and 9am, the time of day in which the most precipitation fell over the city of Amsterdam.

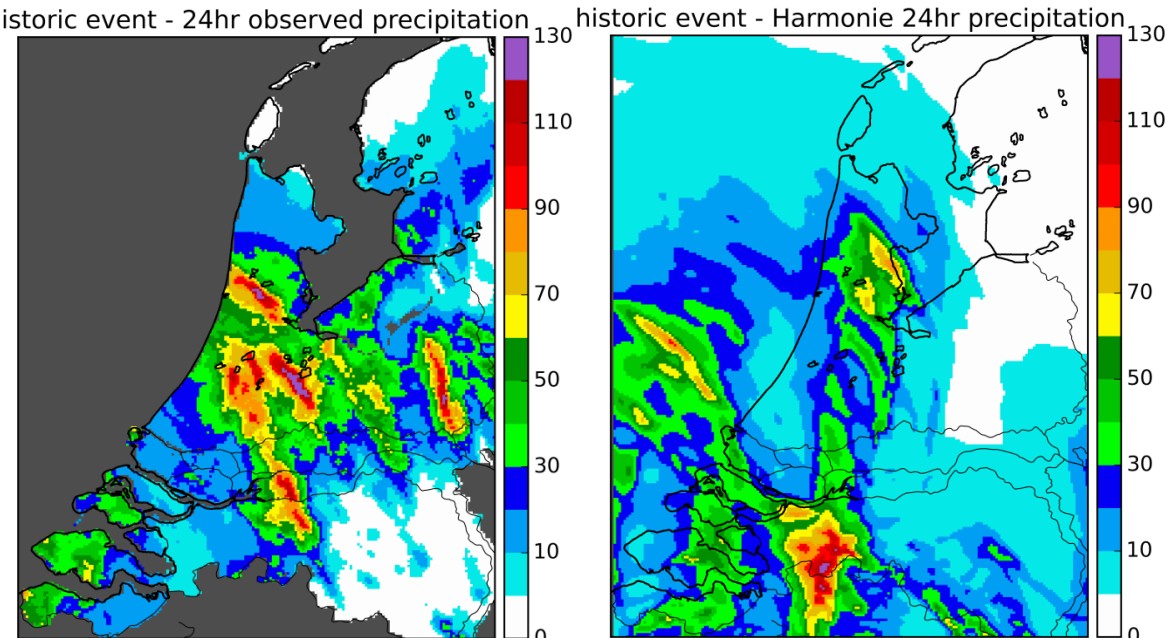

*Figure 2: The daily accumulated precipitation in mm over the Netherlands for July 28, 2014. A) From radar observations (only with available data from the Netherlands' inland regions) and B) As simulated by a representative member from the Harmonie ensemble run.*





## 2.2 The historic event in the Harmonie model

In order to simulate the historic and future events, two ensemble simulations were carried out with the high resolution
weather forecasting model Harmonie (Seity et al., 2011, cycle 40): one under present climate conditions and one under
future climate conditions (2°C warmer). Harmonie uses non-hydrostatic convection permitting dynamics and Arome physics
with a horizontal resolution of 1x1 km$^2$, 60 levels in the vertical direction and a time step of one minute. The output is given
every hour. The initial and boundary conditions are taken from the ECMWF's ENS ensemble runs, are updated every hour
and have a 0.28°x0.28° grid size (~32x32 km$^2$). The ENS is built to predict the probability distribution of forecast states,
taking into account the random analysis error and model uncertainties. In order to select the best-fitted initial and boundary
conditions for the simulation of the present event in Harmonie, the ENS ensemble of 51 members was run for the day of the
event. From the outcome, seven runs that performed closest to the radar observations were selected. These runs initiated the
Harmonie ensemble at 12:00 on July 27, 2014 and ran for 36 hours, rendering an hourly output of the simulated historic
event.This starting time was selected as the precipitable pattern was closer to that of the radar observations, compared to the
runs initiated at 0:00 on July 28.

The outcome of the present ensemble simulation under the initial conditions for July 28 shows that the Harmonie captures
sufficiently well the convective nature, the approximate size of the cells and the maximum intensity of precipitation, as well
as the duration and the approximate time evolution of the event in all of its seven members. However, the location of the
reported events was not accurate (Fig. 2b). Clear discrepancies can be found on the position and number of convective cells
between the simulation and the observations, and between the individual ensemble members (not shown here). The relatively
low predictability of the exact position of the cells is due to the unstable, chaotic character of the specific event and to the
imperfection of the model's initial and boundary conditions.

## 2.3 Scaling Method 1: The Pi-Td from observations

In this section, the methodology for expressing the precipitation intensity as a function of the dew- point temperature is
discussed and compared to CC (Clausius-Clapeyron) scaling. The method was applied to the historic event using a perturbed
input temperature in order to depict the expected intensity changes for a warmer climate.

*The Pi-Td relation*

An important thermodynamic expression for the formation of precipitation in the atmosphere is the CC relation, according to
which the maximum holding capacity of water vapor in an air mass increases by approximately 7% per degree of
warming (Trenberth et al., 2003). When the intensity of heavy precipitation is limited by the local availability of atmospheric
moisture and is not sensitive to the atmospheric dynamic advection processes, it can be expected that the precipitation





intensity increases at the same rate. However, both observations and model simulations show deviations from the CC scaling (Haerter& Berg 2009; Bürger et al., 2014), as the dynamics and feedbacks between the dynamics and the thermodynamics of the atmosphere also play an important role in the formation of precipitation. For example, the relation between extreme precipitation intensity and temperature has been found to reach two times that of the CC scaling, i.e. up to

14% per degree of warming (Lenderink & Van Meijgaard, 2008; Sugiyama et al., 2010; Panthou et al., 2014; Attema et al., 2014; Allan, 2011; Berg et al., 2013). The exceedance of the CC scaling for extreme precipitation is suggested to be related to the large and small-scale dynamics of the atmosphere, and to the vertical stability (Loriaux et al., 2013; Lenderink & Attema 2015). Other studies indicate that statistical factors account for temperature-related changes in precipitation types, with an increasing contribution of convective warmer rain as temperature rises (Haerter & Berg, 2009).

Other processes that potentially play a role are the increase of convective available potential energy (CAPE) with temperature and the positive feedback that is induced by the release of latent heat energy during the condensation of water vapour, thereby enhancing convection (Panthou et al., 2014). In general, the relation between precipitation intensity and temperature varies with region, season, duration and type of precipitation, and is different for low and high temperatures, ranging from below CC to super CC. The scaling can be expressed using either absolute temperature (T) or dew-point

temperature (Td) as a reference. Preference is given to Td, as this quantity contains explicit information on both temperature and the near-surface humidity (Lenderink et al., 2011).

In addition, large-scale circulation, vertical stability, cloud micro-physics, moist adiabatic lapse rate, soil-water scarcity and other factors play a role. The CC or below CC rates are mainly followed by long, synoptic, colder rain, while the super CC is

mainly found in short-lived, warmer convective rain (Panthou et al., 2014; Lenderink et al., 2011; Mishra et al., 2012; Singleton & Toumi, 2013).

The precipitation data used to build the Pi-Td relation is hourly data from the gauge-adjusted Dutch Doppler weather-radar dataset (Overeem et al., 2011). In this data set, the pixel area is approximately 0.9x0.9 km$^2$ and is available for eight years

(2008-2015). The radar operates on the C-band and measures precipitation depths based on composites of reflectivities from two Dutch radar stations: De Bilt and Den Helder. The hourly dew-point temperature was derived from 37 KNMI weather stations in the Netherlands, for the same period as the precipitation data. The sample size of the observed data for the temperature range between 7°C and 21°C can be considered to be large enough to eliminate any statistical artifacts that may occur.


One advantage of the radar's high resolution is that small-scale convective precipitation (~1km) is resolved explicitly. Following Lenderink et al. (2011), only precipitating areas were taken into account (>0.1mm/hr). Rainfall intensity data were first classified into 15 non-uniform percentile classes, ranging from the 25[th] to the 99.9[th] percentile and placed in bins of 2°C Td width overlapping with steps of 1°C. The sensitivity to the temperature bin width was tested by





comparing a 1°C and 0.5°C bin width, and was found to be insignificant. In order to match the precipitation data to antecedent air-mass properties that are characteristic for the formation of the precipitation events, Td was measured four hours prior to the precipitation time. This time shift also avoids the contamination of the temperature and relative-humidity records by the changes that the precipitation process imposes, such as temperature drops due to descending colder, dry

air from convective downdrafts or to heat release from the evaporation of precipitation (Lenderink et al., 2011).

The Pi-Td scaling was calculated separately for low to very high percentiles and is illustrated in Fig. 2. For low to medium temperatures, there is little change of the precipitation intensity with the temperature for all percentiles, while for higher temperatures and percentiles below the medium, a monotonic increase with Td of about 5% per degree is

shown. This behavior is usually attributed to large-scale precipitation and passing synoptic systems. For warmer temperatures between about 15°C-20°C, precipitation intensity increases rapidly with temperature. For medium percentiles the intensity increases at a rate of over 2CC (14% per degree of warming) and rising up to 21% for higher percentiles (a 3CC rate). This rate levels off at very high percentiles and at dew-point temperatures above 21°C, possibly due to a limitation of the moisture supply to sustain the high precipitation intensities, clouds reaching the tropopause, or

statistical artifacts. The extreme 3CC rate is attributed to short-lasting, warm, convective precipitation events. To confirm this, the a comparison to winter conditions was made where the larger synoptic systems are dominant. Also values were computed from daily averaged data, to filter out the effect of short-duration convective events. The rate is almost uniformly CC during the winter, while for daily summer data the rate is below CC for small percentiles and above CC for larger percentiles (not shown). This rapid increase in Pi with Td is also visible in Fig. 1 of Loriaux et al. (2013), where a

3CC rate in the Pi-Td relation is illustrated at the hourly and sub-hourly precipitation, for the 90[th] percentile of a temperature band, similar to the one that is discussed here. For very high percentiles the rate decreases to 2CC.

*The Pi-Td as delta transformation*

A multi-decadal observational analysis in the Netherlands shows that the trend in extreme precipitation can be explained by

changes in dew-point temperatures (Lenderink et al., 2011). In the same study, a similar long-term trend between T and Td indicates an almost constant relative humidity with time, which implies that changes in T scale with changes in Td. Also the KNMI'14 scenarios project no change to a small decrease in the future relative humidity, depending on the scenario.

In the Pi-Td transformation, the dynamics of the atmosphere and the relative humidity are assumed to be unchanged. Starting

from the historic event, the dew-point temperature and precipitation intensity per grid-point are calculated and attributed to a point in the Pi-Td graph (Fig. 3), for the corresponding percentile. The increase in the precipitation intensity, Δpi, is found by moving the initial point along the isolines in Fig. 3 by 2°C. The procedure is repeated for each grid point individually. This method examines only the possible changes in the intensity of precipitation in already precipitable areas and does not allow changes in the spatial scale of the event. The mean dew-point temperature of the event of July 28, 2014 is 17.3°C with a

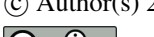



spatial variance of 1.8°C. In addition to the application to the observed records, the Pi-Td method is also applied to the seven members of the simulated historic event shown in Fig 4a below (to be discussed later).

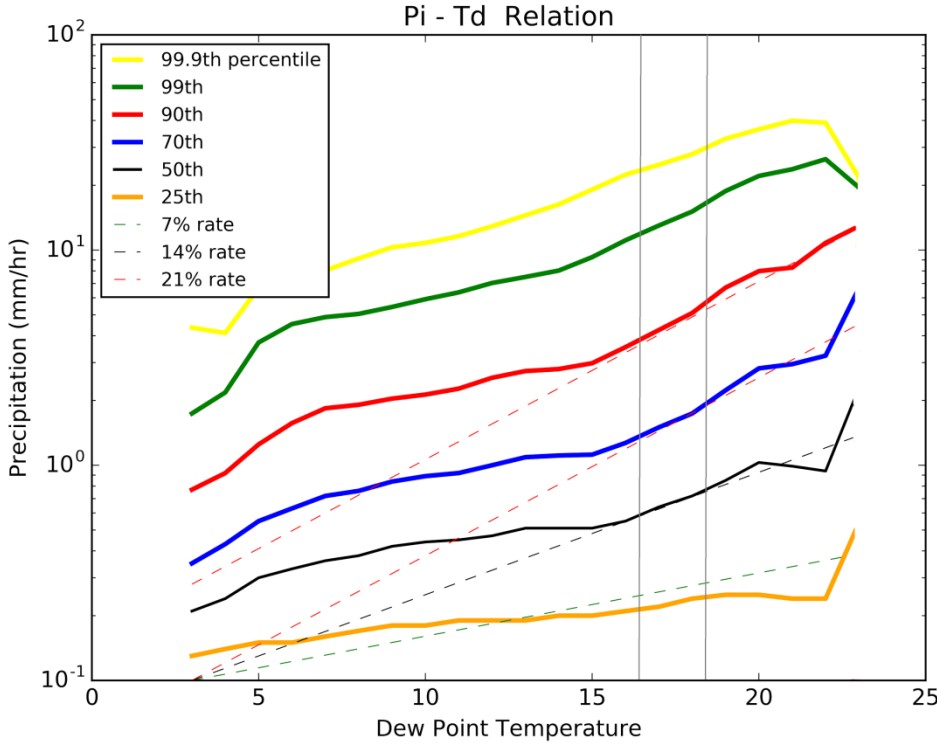

*Figure 3: Precipitation intensity as a function of the dew-point temperature from hourly observations for June-August 2008-2015. The solid lines indicate the different percentiles of precipitation intensity. The dotted lines indicate the 7%/degree, 14% and 21% scaling. The two grey vertical lines indicate the mean temperature on July 28, 2014 in the Netherlands and a 2°C warmer future event.*

## 2.4 Simulating the future weather

For the future event, the Harmonie ensemble was run again, with the temperature of the initial and boundary conditions being increased by 2°C at all levels and time steps. The relative humidity (RH) was kept constant in order to ensure that the provided moisture remained adequate. Due to the constant RH, the temperature change approximately scales with the Td change, resulting in an roughly equal change of 2°C in Td. Attema et al. (2014) show that the simplification of the



homogeneous increase of temperature and RH do not result in significant differences compared to non-homogeneous changes to the temperature and humidity profiles that were derived from a long climate-change simulation.

## 2.5 Scaling Method 2: linear delta transformation from climate models

A common approach to account for climate change effects in hydrological assessments is known as the delta change approach (Andreasson et al., 2004). The change signal between a control (current climate) situation and a future climate condition is used to adjust an observed climate record (such as temperature and precipitation). This adjusted series is subsequently used as input for the hydrological assessment (such as flood simulation). This approach is widely used (see e.g., Hay et al., 2000; Andreasson et al., 2004 and references therein), as it is relatively easy to use and requires only a couple of change factors that can directly be retrieved from either GCM runs or climate scenarios (such as the KNMI'14). Such change factors can differ in terms of complexity, ranging from a single change factor for all values to separate change factors for different months, seasons, and percentiles. In some cases, specific statistical tools have been developed that adjust observed time series by using various parameters that are related to climate change (such as amount of wet days, change in mean, change in extreme) (Bakker & Bessembinder, 2012), as used in Te Linde et al. (2010).

In our case, the linear delta approach is applied with the KNMI'14 scenarios, which are based on the global climate scenarios from the latest IPCC report (Stocker et al., 2014), but tailored to the area of the Netherlands. Four KNMI'14 climate change scenarios were developed for 2050 and 2085. We selected a scenario in which Td is expected to rise by 2°C by 2050. Furthermore, the mean temperature in the selected scenario is expected to increase by 2.3°C and mean summertime RH is expected to decrease by 2.5%. According to this scenario, the maximum hourly intensity of the precipitation per year will increase by a maximum of 25%. In order to up-scale the intensity of the historic event with this linear-delta factor, the entire range of the historic precipitation is increased by 25% (or, assuming a linear increase with temperature, an increase of 11.8% per degree of Td warming).

## 3. Results

## 3.1 The future event from the Pi-Td scaling method

Fig. 4a shows the historic event as simulated by a representative ensemble member in Harmonie, at 9am. The Pi-Td method applied to this event is shown in Fig 4c. As the Pi-Td method only modifies precipitable areas, the future spatial pattern remains unchanged compared to the historic simulated event. In the box-plots of Fig. 5, the intensity increase compared to



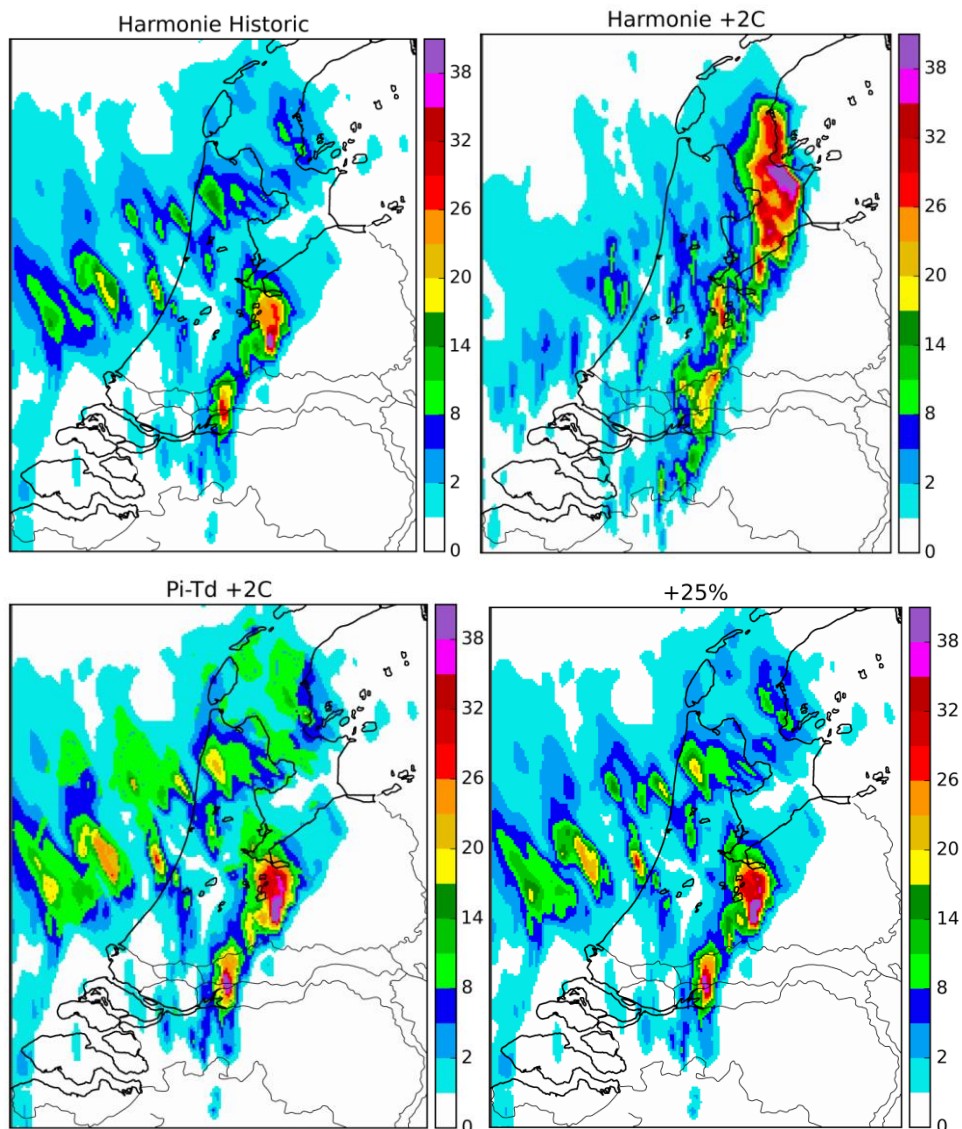

**Figure 4**: *Precipitation intensity shown in Fig. 2b at 9am.A) Harmonie simulation under historic initial and boundary conditions, and B) simulated under future conditions. C) Application of the Pi-Td method and in D) the uniform scaling of +11.8%/°C.*

the simulated historic event is shown for all seven members and for the different precipitation percentiles at 9am and supplementary at 2pm, when the event goes towards its decaying phase. Following the observed scaling of Fig. 2, the lower percentiles (25th) increase with a rate of ΔPi around the CC rate (7%/°C). The medium percentiles (50th) increase between 2CC and over 3CC, and the high percentiles increase from 2CC up to 3CC. The rate of increase decreases slightly for the very high percentiles, reaching a maximum rate of 2CC. There are no considerable differences between the intensity increase





at 9am and 2pm, while some variance is observable between the different members, due to slightly different initial conditions of Pi and Td across the ensemble.

Overall, the Pi-Td method increases the total precipitable water for the entire event by 36%, which is about 17%/°C of warming.

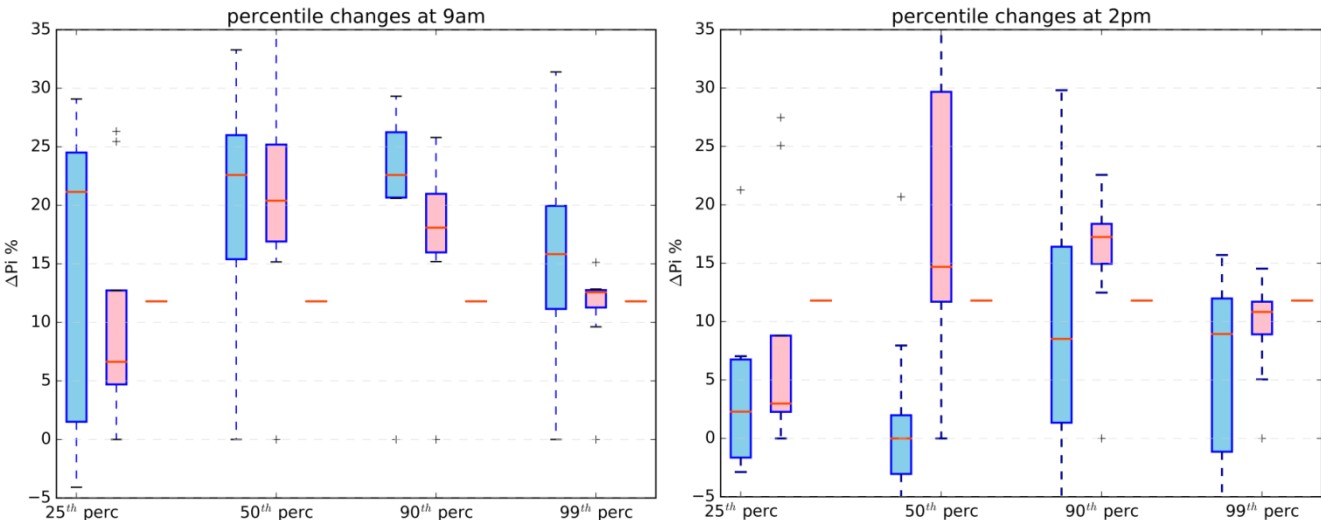

*Figure 5: The change in the hourly precipitation per degree of warming compared to the simulated historic ensemble for the different percentiles and for the three methods over all precipitable points at 9am (top) and 2pm (bottom). The blue box-plots represent the ensemble of the future model run and the pink box-plots represent the outcome of the Pi-Td method, starting from the ensemble of the historic simulation. The single red line indicates the relevant linear transformation of +11.8%/°C.*

## 3.2 The future event in Harmonie

The future event in Harmonie at 9am is shown in Fig. 4b, for the same member as in Fig. 4a. The future event differs in both intensity and precipitable pattern. The maximum Pi is clearly increased and the main body of the precipitable area is shifted towards the northeast in this member, mainly due to changes in horizontal winds. The box-plots that summarize the intensity changes in Fig. 5 show deviations in the response of the model in the morning and in the afternoon. The main Pi increase takes place during the first hours of the event, while the rate of increase later reduces, possibly due to the reduced moisture

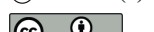

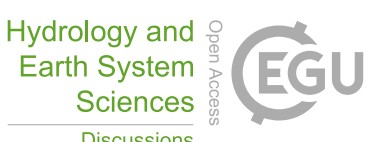

supply that results from the extensive precedent rain. In more detail, the very high percentiles in the morning increase at a rate that lies between 2CC and 3CC, the high percentiles even exceed 3CC and the medium percentiles cover the range of both the high and the very high percentiles. The ΔPi for the lower percentiles varies considerably between the different ensemble members, ranging from a negative ΔPi to a 3CC rate. In the afternoon, the overall rate of increase is substantially

decreased, with an average intensity increase of CC or lower, while some negative values appear in all percentiles.

The variability between the different members primarily results from alterations in the horizontal winds and the convection, due to changes in the surface temperatures, which may shift or change the structure of the clouds. As the event evolves in time, the dynamic heat fluxes and the rapid drying of the soil induce temperature deviations that reach ±4°C locally, thereby

influencing the convection and the horizontal winds.

One interesting outcome is that, despite the temperature increase and the moisture supply, the overall size of the future precipitable domain in all members remains relatively similar to the historic event. A possible explanation could be that, due to the stronger updrafts (caused by extensive warming, and resulting in increased convection and Pi), stronger downdrafts

might be imposed at the outskirts of the clouds, thereby preventing them from expanding further. This may also explain the low or negative scaling that is observable in the low percentiles: as the Pi grows faster spatially within the same domain-size and reaching higher maxima in the future event, there are smaller chances of finding light precipitation.

Overall, the total precipitable water for the entire simulated future event has increased by 27%, which is about 13%/°C of

warming.

### 3.3 The future event with the linear method

The linear delta transformation was applied over the historic simulation of Fig. 4a and the outcome is shown in Fig. 4d. The

precipitation intensity is increased by a total of 25% for a 2°C increase (11.8%/°C). Similarly to the Pi-Td method, there is no change in the spatial domain. Also the variance between the members does not change (box-plots of Fig. 5. The overall duration of the event also remains unchanged compared to the historic event.

### 4. Discussion


All three methods analyzed in this study show an overall increase in the precipitation, together with temperature. Some discrepancies occur in the changes of intensities, duration and the percentile distribution of the future precipitation, as well as in the spatial patterns, the position, and the number and size of the precipitable cells. A summary of the main results is found in table 1.





The fitted lognormal distributions for the frequency of occurrence of the different precipitation intensities (Fig. 6) show strong similarities between the three methods and a clear distinction between present and future. The entire spectrum of the future events is shifted towards higher intensities. The chances of moderate precipitation are reduced and there is a distinct

increase in the frequency of occurrence for Pi>15mm/hr. For example, the probability of the occurrence of intensities higher than 20mm/hr is increased by over 35%. A Kolmogorov-Smirnov test was performed to compare the goodness of fit for various distributions (the beta, gamma, Pareto and lognormal) to conclude that the best fitting distribution for the current data is the lognormal.

Unlike in the Pi-Td and the linear method, the future model simulations show a non-uniform change in Pi with time and space. In the model, the most intense precipitation increase takes place during the first hours, while the rate of increase later drops, possibly due to a drying of the atmosphere resulting from the exceedance of the water that precipitates in the early hours. Harmonie tends to simulate stronger increases in the very high and low precipitation intensities in the first hours of the event, while the Pi-Td method follows a structured and more constant increase that depends only on the Pi and Td of the

historic event at every hour. The total amount of precipitable water that falls in the future Pi-Td event is slightly larger than in the Harmonie future event, due to the model's reduction of the Pi increase in the late hours of the event. The linear method, on the other hand, results in an overall underestimation of the total precipitable amount of water, as it underestimates the Pi increase for the moderate and high percentiles. The duration of the event in the model does not change in the future simulation, in agreement with Chan et al. (2016), in which future simulations with a convective-permitting

model were made to show a clear intensification of sub-hourly rainfall, but no change in rainfall duration.

An intrinsic discrepancy between the model and the delta methods is the ability to shift or build new convective cells, due to the advection of moisture as a result of changes in wind and temperature patterns, which lead to changes in the precipitable spatial patterns. However, as there are hardly any changes in the total precipitable surface area, it can be concluded that the

assumption of a negligible change in the total size of the precipitable area in the delta methods is reasonable for this case study.

It is of interest to investigate whether the different characteristics over sea and land (specifically, the more unified temperatures over sea, the possibility of additional moisture provision and the differences in wind characteristics) could

induce deviations in the behavior of the future event's individual development over sea and over land. However, this experimental setting does not allow for such an analysis, as the spatial domain is rather limited. Changes in the horizontal winds may therefore shift the clouds from over land to over sea or vice-versa, thereby obfuscating the analysis.





Overall, the Pi-Td method appears to render reliable results when compared qualitatively to the model and linear-transformation methods, while it is also faster, less expensive and less complicated. The Pi-Td relation has to be derived explicitly for different locations and different seasons, and is recommended to be used only within the range of well-documented dew-point temperatures for a specific area (e.g., Td>7°C and Td<21°C for the Netherlands in the summer).

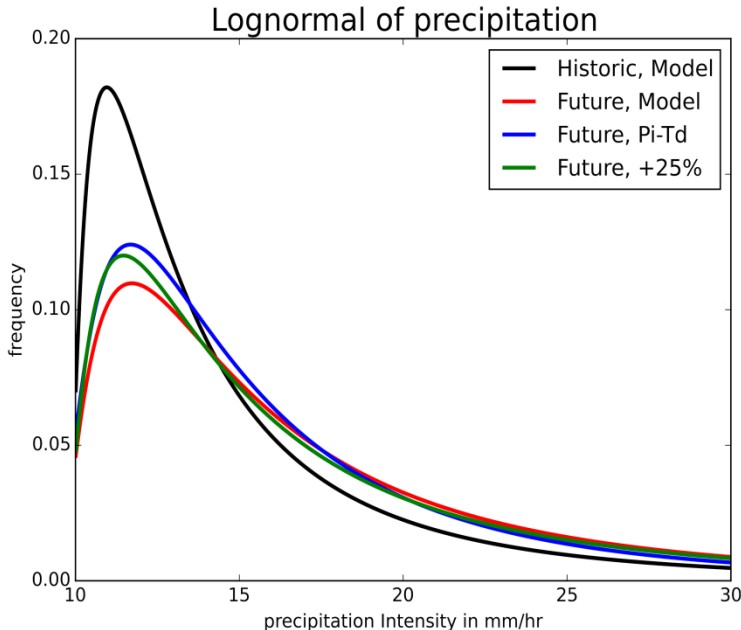

***Figure 6****: The fitted lognormal distribution of precipitation intensities for the present and future events, for all members and methods at 9am.*

|  | Future Pi-Td | Future Harmonie | Future 25% |
|---|---|---|---|
| **Total P** | 17% | 13% | 11.8% |
| **average P** | 17% | 11% | 11.8% |
| **P99.9th** | 12% | 10.5% | 11.8% |
| **Precipitable area** | 0 | 2±3% | 0 |

***Table 1****: Summary of the findings from all ensemble members and at all hours of the event. The numbers show the percentages of change in comparison to the historic ensemble for the three methods and the total precipitation that fell, as well as for the changes in the average intensity, the very extreme percentiles and the precipitable surface area. The noted percentages are per degree of warming, assuming a linear increase.*



## 5. Conclusions

New methods are emerging to project future extreme precipitation as it develops under climate change, grounded in existing
events. For water managers, such future-weather approaches have the advantage that they take a known extreme event as the
basis and simulate its characteristics in a future, warmer climate. However, such an approach requires high-resolution
modelling and can be computationally demanding. In this paper, we compare two novel methods for an historic event in the
Netherlands and one existing-scenario method for projecting future extreme precipitation events starting from historic
events, which can be used for climate research and impact studies.

The first method is a non-linear Pi-Td relation and is used here as a delta-transformation in order to project how a historic
extreme precipitation event would intensify under future, warmer conditions. We show that the hourly summer precipitation
from radar observations with the dew-point temperature (the Pi-Td relation) for moderate to warm days can increase by up to
21% per degree of warming: a relation that is three times higher than the theoretical CC relation. The rate of change depends
on the initial precipitation intensity, whereby low percentiles increase at a rate below CC, the medium and the very high
percentiles (99.9th) at 2CC and the moderate-high and high percentiles at 3CC (90th). In the second method, the future
extreme event is simulated in the Harmonie model, alternating the historic initial conditions to represent a warmer
atmosphere. Finally, the third method applies a linear delta transformation over the simulated historic event. The linear delta
arises from the KNMI'14 scenarios, according to which all precipitation percentiles experience an increase of 11.8%/°C in
their intensities.

The comparison between the three future-weather methods shows a comparable increase in the precipitation intensities,
which range from below CC to a 3CC rate of change per degree of warming, depending on the initial percentiles. Some
divergence is found in the distribution of the intensity changes, the time evolution of the event and the position of the
precipitable cells, due to the intrinsic discrepancies between the methods.

While the Pi-Td method focuses primarily on the contribution of the thermodynamics and statistics in order to conclude on
the behavior of the precipitation with temperature, the future-weather method in Harmonie explores both the atmospheric
dynamics and the thermodynamics, as well as on their interactions. Each run can evolve differently with time, while
resolving the complicated atmospheric dynamics may increase the noise in the outcome.

A noteworthy discrepancy is that, in the Harmonie model, the intensity changes are not uniform with time, as the main Pi
increase takes place during the first hours of the event, while the rate of increase later reduces, possibly due to an exhaustion
of atmospheric moisture resulting from the extensive precedent rain. Overall, the total increase in the precipitable amount of



water was increased by about 13%/°C in the model method, 17%/°C in the Pi-Td method and 11.8%/°C in the linear method. Due to small wind and convection changes in the model, the clouds' position and patterns are displaced. Nevertheless, in the model, total-spatial precipitable coverage remains practically unchanged with temperature change, as is also assumed in the statistical methods.

The Pi-Td method also has limitations, as it focuses on the precipitation-intensity changes, while it does not answer questions on spatial distribution, time evolution and return-period changes, or about changes in the synoptic state of the atmosphere. For example, it is suggested that in the future rate of precipitation intensities with temperature may decline over the UK, due to the more frequent occurrence of anticyclonic systems (Chan et al., 2016), indicating that there is a possibility

for some change in the future Pi-Td scaling, in some places.

The Pi-Td method projects precipitation events at different temperatures and is simple to use, requires little time and is computationally and resource efficient, while it continues to offer rather robust results compared to a relevant non-hydrostatic model simulation. In all cases, the variance of the results with the Pi-Td method is smaller than with the model

method, allowing for a more straightforward and deterministic analysis if the outcome is to be used for impact studies. This method is suggested for use within well-documented temperature ranges deriving from observations in order to avoid statistical artifacts in the Pi-Td scaling. Therefore it is not recommended to be used for very high (or very low) temperatures.

The outcome of the Pi-Td future event can be used in several applications, such as impact and risk analyses by which to

assess the economic and environmental damages of a future extreme event over an urban (or rural, industrial) area, thereby allowing policy makers to evaluate adequate adaptation measures against future disasters. It can also be used in several regional hydrological or larger spatial-scale climatological studies.

**Acknowledgements**

This research has been funded by Amsterdam Water Science AAA and the NWO-VICI grant 453-13-006. We would like to thank KNMI for their support to offer access to the Harmonie model.

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
