# Peer review of "Future extreme precipitation intensities based on a historic event"

_Hydrology and Earth System Sciences, 2017_

## Referee Comment (RC1) · Anonymous Referee #1 · 25 Jul 2017

The article will be acceptable, although some revision is desired to make it easier to understand.

[Specific comments] (1) The authors describe a number of approaches and methods to estimate future extreme events in Chapters 1 and 2. It is desired to make their relations clearer for readers who do not know much about techniques in this field.

@ Please make clear how the three methods mentioned in the upper part of Page 3 (future-weather method, first scaling method, and second scaling method) are related to the "current approaches" described in the middle part of Page 2.

@ Also, please make clear how these three methods correspond to the three methods that appear in the first paragraph of Chapter 2 (Pi-Td scaling method, future weather

method, and linear delta-change method).

@ It is better to write the method name in each box in Fig.1.

(2) It will be better to change the phrase "historic events" in the Title to "a historic event", because the study was made for a single case.

[Technical comments] @ Line 30 on Page 2 "Selection of events that have triggered concerns by for instance flood-risk managers using leads to —": I cannot understand the sentence. Possibly the word "using" is unnecessary.

@ It is better to show the location of Amsterdam on a map.

@ Line 9 on Page 6 "0.28deg x 0.28deg grid size (∼32x32 km2)": 0.28 deg in longitude will be about 20 km at the latitude of 50 deg.

@ Line 7 on Page 8 "Fig.2": Fig.3

@ Line 20 on Page 10 "Td is expected to rise by 2 deg by 2050. — the entire range of the historic precipitation is increased by 25% (or, assuming a linear increase with temperature, an increase of 11.8% per degree of Td warming)." 11.8% x 2deg is 23.6%, which is different from 25%. Please check.

@ It is better to write the number of members for Fig.5 (seven?).

@ Line 11 on Page 12 "9am (top) and 2pm (bottom).": left and right.

@ Line 4 on Page 12 "the Pi-Td method increases the total precipitable water for the entire event by 36%, which is about 17%/deg of warming.": If the increase of Td is 2 deg, then the increase of 36% means 18%/deg of warming. Please check. The situation is similar for the phrase "27%, which is about 13%/deg" in Line 19 on Page 13.

@ Line 9 on Page 13 "the rapid drying of the soil": Why does the soil dry after precipitation?

---

## Referee Comment (RC2) · Anonymous Referee #2 · 20 Oct 2017

Many studies purport the best way to project rainfall for a future climate. In this study an honest and unambiguous assessment is presented comparing different methods of storm projection. As far as I am aware, this is the first study of its kind, and will be of interest to both the research community and practioners alike. The commensurable results are very promising, and the uncertainty in the results presents a need for a greater understanding in this field. This is a very worthwhile contribution.

My suggestions are very minor and primarily focus on expanding the literature cited and ensuring all assumptions have been documented. I look forward to seeing the published manuscript.

Minor comments:

[Figure]

\# I think you need a reference or two and a sentence on the assumption of constant humidity. I have no doubt that this assumption is fine but on Page 3, Line 32 you state you assume constant humidity and then in Line 21 of Page 10 you state humidity is expected to change. I would insert a sentence or two on the predicted changes of humidity on Page 3 and reference accordingly (say from the IPCC reports) so the reader can then make an assessment of the validity of this assumption. I stress that this assumption is valid – it just needs to be communicated.

\# Section 3 would read better without the subheadings. It currently feels a little disjointed and repetitive. The reason is that you start by comparing the results of the Pi-Td scaling to the Harmonie model and then repeat the presentation of the Harmonie results in Section 3.2. This could all be synthesised into one section. Presenting all the panels up front in Section 3 would read better and grouping the results for the overall precipitation intensity in one paragraph would also read better.

\# I think somewhere in the discussion or conclusion the fact that storms may change in their duration/type/frequency should be acknowledged as something that isn't considered here e.g. Molnar et al., (2015).

\# The conclusion (and assumption) of non-changing spatial patterns/size needs to be discussed with in line with the current literature. See Guinard et al., (2015); Wasko et al., (2016) ; Lochbihler et al., (2017). This will help strengthen the findings presented here.

\# None of the figures have the panels labelled (e.g. a, b, c, d) – Figure 5 is not top/bottom. Line by line comments:

\# Page 1, Line 20: The line break isn't needed.

\# Page 1, Line 31: Changing antecedent conditions is also important to understand in this context and should be acknowledged, e.g Ivancic and Shaw (2015) and Wasko and Sharma (2017).

**Page 2, Line 20: I would cite Fowler et al (2007) here.**

**Page 3, Line 9: I think Lenderink and Attema (2015) needs to be cited alongside this reference.**

**Page 5, Line 1: A reference to these changes in storms would be beneficial here.**

**Page 7: Line 5: Global studies could be cited here, see the following papers: 10.1029/2011GL048426; 10.1002/2016GL071354**

**Page 7, Line 9 and Line 21: Both Molnar et al., (2015) and Wasko et al., (2015) show different types of artefacts related to increased short duration convective rainfall at higher temperatures resulting in higher scaling.**

**Page 7, Line 28: The statement that the sample size is large is vague – maybe state the number. Also state explicitly that all precipitation pixels were used. I couldn't tell from the text but I assume this is the case.**

**Page 8, Line 5: Another manuscript which comments on this explicitly is Bao et al (2017)**

**Page 8, Line 15: I think you need a reference on the statistical artefacts – one such paper is Wasko et al., (2015) which relates to embedded storms, another is Molnar et al., (2015) relates to mixing of storms. Also Hardwick-Jones et al (2010) is usually cited in relation to moisture limitations.**

**Page 8, Line 17: remove "the"**

**Page 8, Line 25: Again also cite Lenderink and Attema (2015).**

**Page 10, Line 32: The unchanged spatial pattern is also true for the delta change method – could be stated here.**

**Page 14, Line 14: Around here a reference back to Figure 5 would be beneficial.**

References:

Bao, J., S. C. Sherwood, L. V. Alexander, and J. P. Evans (2017), Future increases in extreme precipitation exceed observed scaling rates, Nat. Clim. Chang., 7(2), 128–132, doi:10.1038/nclimate3201.

Fowler, H. J., S. Blenkinsop, and C. Tebaldi (2007), Linking climate change modelling to impacts studies: recent advances in downscaling techniques for hydrological modelling, Int. J. Climatol., 27(12), 1547–1578, doi:10.1002/joc.1556.

Guinard, K., A. Mailhot, and D. Caya (2015), Projected changes in characteristics of precipitation spatial structures over North America, Int. J. Climatol., 35(4), 596–612, doi:10.1002/joc.4006.

Hardwick Jones, R., S. Westra, and A. Sharma (2010), Observed relationships between extreme sub-daily precipitation, surface temperature, and relative humidity, Geophys. Res. Lett., 37(22), L22805, doi:10.1029/2010GL045081.

Ivancic, T. J., and S. B. Shaw (2015), Examining why trends in very heavy precipitation should not be mistaken for trends in very high river discharge, Clim. Change, 133(4), 681–693, doi:10.1007/s10584-015-1476-1.

Lochbihler, K., G. Lenderink, and A. P. Siebesma (2017), The spatial extent of rainfall events and its relation to precipitation scaling, Geophys. Res. Lett., 44(16), 8629–8636, doi:10.1002/2017GL074857.

Molnar, P., S. Fatichi, L. Gaál, J. Szolgay, and P. Burlando (2015), Storm type effects on super Clausius–Clapeyron scaling of intense rainstorm properties with air temperature, Hydrol. Earth Syst. Sci., 19(4), 1753–1766, doi:10.5194/hess-19-1753-2015.

Wasko, C., A. Sharma, and F. Johnson (2015), Does storm duration modulate the extreme precipitation-temperature scaling relationship?, Geophys. Res. Lett., 42(20), 8783–8790, doi:10.1002/2015GL066274.

Wasko, C., A. Sharma, and S. Westra (2016), Reduced spatial extent of extreme storms at higher temperatures, Geophys. Res. Lett., 43(8), 4026–4032,

doi:10.1002/2016GL068509.

Wasko, C., and A. Sharma (2017), Global assessment of flood and storm extremes with increased temperatures, Sci. Rep., 7(1), 7945, doi:10.1038/s41598-017-08481-1.

---

## Author Comment (AC1) · 24 Nov 2017

We would like to thank the reviewer for her/his time and constructive comments. We have revised the document accordingly and addressed each of the comments and suggestions in their response.

The article will be acceptable, although some revision is desired to make it easier to understand. [Specific comments] The authors describe a number of approaches and methods to estimate future extreme events in Chapters 1 and 2. It is desired to make their relations clearer for readers who do not know much about techniques in this field.

@ Please make clear how the three methods mentioned in the upper part of Page 3 (future-weather method, first scaling method, and second scaling method) are related

to the "current approaches" described in the middle part of Page 2. In the introduction section of the manuscript it is mentioned that in order to provide accurate information on how extreme weather events may look like in the future, sufficiently long and precise model simulations would be necessary. Since those are not yet existent, alternatives are being employed. Some of those are being listed as broad categories: Method A) the delta change technique (or elsewhere named 'delta transformation'), which is a statistical approach that transforms observed data based on changes found in simulated data, Method B) downscaling techniques, Method C) bias-correction techniques, and Method D) the future weather method, in which an observed event is simulated in a high resolution model to show how the same event would occur in a future, warmer climate.

In this study three techniques are further explained and implemented. The two of them belong to Method A and the third to Method D. The first is a scaling method, based on the non-linear delta transformation Pi-Td (Method A), and the second is a simplistic linear delta-change technique (also Method A). These methods will be referred to as 'delta transformation' and the 'simple linear delta transformation', respectively. The model method belongs to the Method D, the 'future weather method'. This explanation is now explicitly mentioned in the introduction of the revised paper. In order to further avoid confusion.

@ Also, please make clear how these three methods correspond to the three methods that appear in the first paragraph of Chapter 2 (Pi-Td scaling method, future method, and linear delta-change method).

It will be clearly mentioned in the text of chapter 2 that 1) the Pi-Td scaling method refers to the first scaling method in the introduction that follows a non-linear delta transformation (Method A), 2) the linear delta-change (or delta transformation) refers to the simplistic linear delta change in the introduction (also Method A) and 3) the future weather method (Method D).

@ It is better to write the method name in each box in Fig.1.

To further clarify the names and uses of the 3 different methods the names, as suggested by the reviewer, will be mentioned in the flowchart as seen in the Figure 1, attached below.

It will be better to change the phrase "historic events" in the Title to "a historic event", because the study was made for a single case. We agree and will change the title to 'a historic event'.

[Technical comments]

@ Line 30 on Page 2 "Selection of events that have triggered concerns by for instance flood-risk managers using leads to —": I cannot understand the sentence. Possibly the word "using" is unnecessary.

The sentence is corrected to: "By basing a future situation on past events that are known to flood-risk managers, it is much easier for them to interpret the impact of such hypothetical future conditions. "

@ It is better to show the location of Amsterdam on a map.

Amsterdam is now shown using a black box in figure 2.

@ Line 9 on Page 6 "0.28deg x 0.28deg grid size (32x32 km2)": 0.28 deg in longitude will be about 20 km at the latitude of 50 deg.

This is true, thank you for the notification. The size should indeed be 32x20km2.

@ Line 7 on Page 8 "Fig.2": Fig.3

This is now corrected.

@ Line 20 on Page 10 "Td is expected to rise by 2 deg by 2050. — the entire range of the historic precipitation is increased by

25% (or, assuming a linear increase with temperature, an increase of 11.8% per degree

of Td warming)." 11.8% x 2deg is 23.6%, which is different from 25%. Please check. Thank you for the notification, the text should be corrected to "steady increase with temperature" (instead of linear). That is 11.8% increase per degree, therefore for 2 degrees would be 11.8%*11.8% = 25% increase.

@ It is better to write the number of members for Fig.5 (seven?).

In the caption of figure 5 we now mention that there are seven ensemble members.

@ Line 11 on Page 12 "9am (top) and 2pm (bottom).": left and right.

Indeed, is now corrected.

@ Line 4 on Page 12 "the Pi-Td method increases the total precipitable water for the entire event by 36%, which is about 17%/deg of warming.": If the increase of Td is 2 deg, then the increase of 36% means 18%/deg of warming. Please check.

The situation is similar for the phrase "27%, which is about 13%/deg" in Line 19 on Page 13. By a total 36% of warming with a steady increase for 2C we mean that x%*x% = 36%, so x=16.62%, or rounded for simplicity to a 17%. Similarly, the 27% increase for 2 degrees would result from a 12.7% per degree steady increase, which is rounded in the text of the manuscript to a 13% per degree. We will clarify this in the text of the manuscript in line 23 of page 10, where this example of calculation is made for the first time in the text.

@ Line 9 on Page 13 "the rapid drying of the soil": Why does the soil dry after precipitation?

Treatment of the land surface in Harmonie model is a compromise between physical accuracy and availability of local information. Previous (non-documented) experience with the land surface module in Harmonie has shown that the representation of both vegetation (controlling evaporation) and soil hydraulics (controlling drainage to deeper layers) is not very well adjusted to local conditions in many locations of the simulation domain. Particularly under extreme conditions - such as the case explored in this

study - this may lead to imperfect representation of the relevant processes such as the dynamics of soil water in the land surface module.

[Figure]

3 methods of projecting the event to future conditions

Present Event

Build the **Pi-Td delta transformation** from observations and apply on present event

Apply the **future weather concept**: simulate the event under warmer conditions in Harmonie model

Apply a **linear delta transformation** from models of KNMI'14 scenarios on present event

Future Event 1

Future Event 2

Future Event 3

Comparison of methods

**Fig. 1.**

---

## Author Comment (AC2) · 24 Nov 2017

We thank the reviewer for her/his valuable comments and useful citation suggestions. We have revised the document accordingly and addressed each of the comments and suggestions in this response.

Many studies purport the best way to project rainfall for a future climate. In this study an honest and unambiguous assessment is presented comparing different methods of storm projection. As far as I am aware, this is the first study of its kind, and will be of interest to both the research community and practioners alike. The commensurable results are very promising, and the uncertainty in the results presents a need for a greater understanding in this field. This is a very worthwhile contribution. My suggestions are

very minor and primarily focus on expanding the literature cited and ensuring all assumptions have been documented. I look forward to seeing the published manuscript. Minor comments:

**I think you need a reference or two and a sentence on the assumption of constant humidity. I have no doubt that this assumption is fine but on Page 3, Line 32 you state you assume constant humidity and then in Line 21 of Page 10 you state humidity is expected to change. I would insert a sentence or two on the predicted changes of humidity on Page 3 and reference accordingly (say from the IPCC reports) so the reader can then make an assessment of the validity of this assumption. I stress that this assumption is valid – it just needs to be communicated.**

We thank the reviewer for this comment. In the initial manuscript we state this assumption on page 8 lines 24-27: "A multi-decadal observational analysis in the Netherlands shows that the trend in extreme precipitation can be explained by changes in dew-point temperatures (Lenderink et al., 2011). In the same study, a similar long-term trend between T and Td indicates an almost constant relative humidity with time, which implies that changes in T scale with changes in Td. Also the KNMI'14 scenarios project no change to a small decrease in the future relative humidity, depending on the scenario." We will include these two references in page3, line 32 to make clear that it is a valid assumption.

**Section 3 would read better without the subheadings. It currently feels a little disjointed and repetitive. The reason is that you start by comparing the results of the Pi-Td scaling to the Harmonie model and then repeat the presentation of the Harmonie results in Section 3.2. This could all be synthesised into one section. Presenting all the panels up front in Section 3 would read better and grouping the results for the overall precipitation intensity in one paragraph would also read better.**

We agree and have now merged the different results section to avoid repetition. The new results section would then read as follows: " 3. Results

Fig. 4a shows the historic event as simulated by a representative ensemble member in Harmonie, at 9am. The relevant future event for the same member as simulated by Harmonie model is shown in Fig. 4b, as resulted by the Pi-Td method in Fig. 4c and by the linear delta transformation in Fig. 4d. It is shown that the maximum Pi is clearly increased in all three methods. As the Pi-Td and linear delta methods only modify precipitable areas, the future spatial pattern remain unchanged compared to the historic simulated event. Conversely, the simulated future event differs in both intensity and precipitable pattern. The main body of the precipitable area is shifted towards the northeast in this member, mainly due to changes in horizontal winds. The variability between the different members primarily results from alterations in the horizontal winds and the convection, due to changes in the surface temperatures, which may shift or change the structure of the clouds. As the event evolves in time, the dynamic heat fluxes and the rapid drying of the soil induce temperature deviations that reach $\pm 4°$C locally, thereby influencing the convection and the horizontal winds. One interesting outcome in the simulated future weather method is that, despite the temperature increase and the moisture supply, the overall size of the future precipitable domain in all members remains relatively similar to the historic event. A possible explanation could be that, due to the stronger updrafts (caused by extensive warming, and resulting in increased convection and Pi), stronger downdrafts might be imposed at the outskirts of the clouds, thereby preventing them from expanding further. This may also explain the low or negative scaling that is observable in the low percentiles: as the Pi grows faster spatially within the same domain-size and reaching higher maxima in the future event, there are smaller chances of finding light precipitation. The box-plots of Fig. 5 depict the intensity increase of the three methods compared to the simulated historic event for all seven members and for various precipitation percentiles at 9am and supplementary at 2pm, when the event goes towards its decaying phase. In the Pi-Td method, following the observed scaling of Fig. 2, the lower percentiles (25th) increase with a rate of $\Delta$Pi around the CC rate (7%/$°$C). The medium percentiles (50th) increase between 2CC and over 3CC, and the high percentiles increase from 2CC up to 3CC. The rate of

increase decreases slightly for the very high percentiles, reaching a maximum rate of 2CC. There are no considerable differences between the intensity increase at 9am and 2pm, while some variance is observable between the different members, due to slightly different initial conditions of Pi and Td across the ensemble. In the linear delta method the increase is a constant 11.8%/°C with no the variance between the members. The overall duration of the event in both Pi-Td and linear delta remain unchanged compared to the historic event. On the other hand, the simulated future weather method in Harmonie in Fig. 5 shows deviations in the response of the model in the morning and in the afternoon. The main Pi increase takes place during the first hours of the event, while the rate of increase later reduces, possibly due to the reduced moisture supply that results from the extensive precedent rain. In more detail, the very high percentiles in the morning increase at a rate that lies between 2CC and 3CC, the high percentiles even exceed 3CC and the medium percentiles cover the range of both the high and the very high percentiles. The $\Delta$Pi for the lower percentiles varies considerably between the different ensemble members, ranging from a negative $\Delta$Pi to a 3CC rate. In the afternoon, the overall rate of increase is substantially decreased, with an average intensity increase of CC or lower, while some negative values appear in all percentiles. Overall, the total increase in the precipitable water for the entire event duration for a 2°C of warming in the Pi-Td method is 36%, which is about 17%/°C, the total increase in the future weather method is 27% (or 13%/°C) and the total increase in the linear delta transformation is 25% (or 11.8%/°C). "

\# I think somewhere in the discussion or conclusion the fact that storms may change in their duration/type/frequency should be acknowledged as something that isn't considered here e.g. Molnar et al., (2015). We touched upon this briefly in the conclusions section, page 17, lines 5-10. This section is now enriched with the conclusions from Molnar et al. 2015 as follows:

"The Pi-Td method also has limitations, as it focuses on the precipitation-intensity changes, while it does not answer questions on spatial distribution and time evolution. Different precipitation types may also show different precipitation behavior with the temperature increase, as seen in Molnar et al. (2015), where observations showed that the intensity increase with temperaturein convective events is higher than that of the synoptic storms. It should be stated that none of the three methods include information on changes in return-period of events, or changes in the synoptic state of the atmosphere. For example, it is suggested that in the future rate of precipitation, intensities with temperature may decline over the UK, due to the more frequent occurrence of anticyclonic systems (Chan et al., 2016), indicating that there is a possibility for some change in the future Pi-Td scaling, in some places.

**The conclusion (and assumption) of non-changing spatial patterns/size needs to be discussed with in line with the current literature. See Guinard et al., (2015); Wasko et al., (2016) ; Lochbihler et al., (2017). This will help strengthen the findings presented here.**

The relevant text is now revised as follows:

"Nevertheless, in the model, the total precipitable coverage remains practically unchanged with temperature change, as is also assumed in the two statistical methods. This case study finding might be contradicting with the recent observational study of Lochbihler et al., (2017), where Dutch radar precipitation data were used, to conclude that on average the precipitable cells increase with increasing temperature and precipitation intensity, especially at higher dew point temperatures. On the other hand, Wasko et al. (2016) found evidence that precipitation intensity in Australia increases with temperature, while the storm's spatial extent decreases, as a redistribution of moisture toward the center takes place at the cost of the outer region of the precipitable area. The model study of Guinard et al. (2015) supports that the changes in precipitable structures with temperature are sensitive to the climatic region and the season."

**None of the figures have the panels labelled (e.g. a, b, c, d) – Figure 5 is nottop/bottom.**

This is now fixed.

Line by line comments:

**Page 1, Line 20: The line break isn't needed.**

Is now corrected.

**Page 1, Line 31: Changing antecedent conditions is also important to understandin this context and should be acknowledged, e.g Ivancic and Shaw (2015) and Wasko and Sharma (2017).**

These citations are included in line 24 of page 1 as follows:

"Different types of flooding may result from extreme precipitation, while the antecedent soil conditions also play a role on stream discharge levels (Ivancic and Shaw (2015) and Wasko and Sharma (2017)"

**Page 2, Line 20: I would cite Fowler et al (2007) here.**

Fowler et al. 2007 citation is added at this line.

**Page 3, Line 9: I think Lenderink and Attema (2015) needs to be cited alongside this reference.**

Lenderink and Attema (2015) citation is added at this line.

**Page 5, Line 1: A reference to these changes in storms would be beneficial here.**

The relevant citation for these findings is KNMI'14, now stated more clearly in the text.

**Page 7: Line 5: Global studies could be cited here, see the following papers: 10.1029/2011GL048426; 10.1002/2016GL071354**

The citations are included and the text is modified as follows:

"For example, the relation between extreme precipitation intensity and temperature has been found to reach two times that of the CC scaling, i.e. up to 14% per degree

of warming (Lenderink& Van Meijgaard, 2008; Sugiyama et al., 2010; Panthou et al., 2014; Attema et al., 2014; Allan, 2011; Berg et al., 2013). This scaling relation shows someÂălarge spatial inhomogeneity (Wasko et al., 2016), with the strong scaling found mainly in the mid- and high latitudes, while in the tropics extreme precipitation intensities are found to exhibit even a decrease with increasing dew point temperatures (Utsumi et al., 2011)."

**Page 7, Line 9 and Line 21: Both Molnar et al., (2015) and Wasko et al., (2015) show different types of artefacts related to increased short duration convective rainfall at higher temperatures resulting in higher scaling.**

Those citations are now added in the text.

**Page 7, Line 28: The statement that the sample size is large is vague – maybe state the number. Also state explicitly that all precipitation pixels were used. I couldn't tell from the text but I assume this is the case.**

This temperature range includes 97% of 8 years of hourly summer data of 1x1km2 resolution for the Netherlands. Indeed, all precipitation pixels are used. This is stated now more clearly in the text.

**Page 8, Line 5: Another manuscript which comments on this explicitly is Bao et al (2017).**

This citation is now included.

**Page 8, Line 15: I think you need a reference on the statistical artefacts – one such paper is Wasko et al., (2015) which relates to embedded storms, another is Molnar et al., (2015) relates to mixing of storms. Also Hardwick-Jones et al (2010) is usually cited in relation to moisture limitations.**

The citations of Wasko et al., (2015) and Hardwick-Jones et al (2010) are now added in the text. The statistical artefacts referred at this sentence refer to the levelling off of the CC scaling, while Molnar et al. attributes an observed increase in this scaling

to the mixing types of the storms. We believe that this citation fits better in the "Pi-Td relation" section and is suggested ti be added there.

**Page 8, Line 17: remove "the"**

The "the" is now removed.

**Page 8, Line 25: Again also cite Lenderink and Attema (2015).**

Lenderink and Attema (2015) citation is added at this line.

**Page 10, Line 32: The unchanged spatial pattern is also true for the delta changemethod – could be stated here.**

The results section is re-arranged, as seen the second point of this revision and this statement is now more clearly put for the reader.

**Page 14, Line 14: Around here a reference back to Figure 5 would be beneficial.**

A reference to Fig. 5 is now made to make this point of the discussion more comprehensive.

References:

Bao, J., S. C. Sherwood, L. V. Alexander, and J. P. Evans (2017), Future increases in extreme precipitation exceed observed scaling rates, Nat. Clim. Chang., 7(2), 128– 132, doi:10.1038/nclimate3201. Fowler, H. J., S. Blenkinsop, and C. Tebaldi (2007), Linking climate change modelling to impacts studies: recent advances in downscaling techniques for hydrological modelling, Int. J. Climatol., 27(12), 1547–1578, doi:10.1002/joc.1556. Guinard, K., A. Mailhot, and D. Caya (2015), Projected changes in characteristics of precipitation spatial structures over North America, Int. J. Climatol., 35(4), 596–612, doi:10.1002/joc.4006. Hardwick Jones, R., S. Westra, and A. Sharma (2010), Observed relationships between extreme sub-daily precipitation, surface temperature, and relative humidity, Geophys. Res. Lett., 37(22), L22805, doi:10.1029/2010GL045081. Ivancic, T. J., and S. B. Shaw (2015), Examining why

trends in very heavy precipitation should not be mistaken for trends in very high river discharge, Clim. Change, 133(4), 681–693, doi:10.1007/s10584-015-1476-1. Lochbihler, K., G. Lenderink, and A. P. Siebesma (2017), The spatial extent of rainfall events and its relation to precipitation scaling, Geophys. Res. Lett., 44(16), 8629– 8636, doi:10.1002/2017GL074857. Molnar, P., S. Fatichi, L. Gaál, J. Szolgay, and P. Burlando (2015), Storm type effects on super Clausius–Clapeyron scaling of intense rainstorm properties with air temperature, Hydrol. Earth Syst. Sci., 19(4), 1753–1766, doi:10.5194/hess-19-1753-2015. Wasko, C., A. Sharma, and F. Johnson (2015), Does storm duration modulate the extreme precipitation-temperature scaling relationship?, Geophys. Res. Lett., 42(20), 8783–8790, doi:10.1002/2015GL066274. Wasko, C., A. Sharma, and S. Westra (2016), Reduced spatial extent of extreme storms at higher temperatures, Geophys. Res. Lett., 43(8), 4026–4032. Wasko, C., and A. Sharma (2017), Global assessment of flood and storm extremes with increased temperatures, Sci. Rep., 7(1), 7945, doi:10.1038/s41598-017-08481-1.

---

## Author Response (AR2)

Dear Carlo De Michele,

Please find attached the revised manuscript of the work "Future extreme precipitation events based on a historic event".
Below you can find the replies to the final reviewer. The comments of the reviewer are provided in *Italic* fonts and are followed by our answers.

I am looking forward to your reply.

Best wishes,
Iris Manola

We would like to thank the reviewer for her/his time and constructive comments. We have revised the document accordingly and addressed each of the comments and suggestions in their response.

**REVIEWER #1**
*I appreciate the authors' effort of revision. The manuscript is almost ready for acceptance. However, some additional explanation will make the article easier to understand (although revision is not mandatory).*

*(1) The authors have presented following methods in the Introduction as written in their reply to my previous comments:*

*@ Method A: the delta change technique,*
*@ Method B: downscaling techniques,*
*@ Method C: bias-correction techniques, and*
*@ Method D: the future weather method.*

*On the other hand, the authors' analysis is based on two types of application of "A" and an application of "D". It will be better to write explicitly that B and C are not used in the present study.*

Answer:
The text in page 3, lines 14-16 are revised as follows in order to clarify that only the two out of the four methods are utilized in the paper:

"Among the four methods described above (delta change, downscaling techniques, bias correction and future weather), the delta change and the future weather are employed in this paper. The main

aim is to compare a 'future weather' simulation with two alternative 'delta-change' scaling methods, of which one is developed in this study."

*(2) It appears to me that the "future weather" method is the same as the "pseudo global warming" method that is widely used for climate change studies (e.g., doi:10.1175/JCLI-D-15-0623.1, 10.1175/JCLI-D-16-0697.1). If I understand correctly, it will be better to write so for the convenience of readers.*

Thank you for this notification. The following text is added in page 3, line 8:

"A similar method is the 'pseudo global warming' method, which involves the simulation of observed events modifying the meteorological forcing by a climate change difference (Schär et al. 1996, Michaelis et al., 2017). For example, Trapp & Hoogewind 2016 applied climate change differences from CMIP5 simulations on the high resolution Weather Research and Forecasting (WRF) Model to reveal how typical extreme observed tornadoes might be realized under conditions of the late twenty-first century."